# Piezo-Responsive Hydrogen-Bonded Frameworks Based on Vanillin-Barbiturate Conjugates

**DOI:** 10.3390/molecules27175659

**Published:** 2022-09-02

**Authors:** Anna S. Nebalueva, Alexandra A. Timralieva, Roman V. Sadovnichii, Alexander S. Novikov, Mikhail V. Zhukov, Aleksandr S. Aglikov, Anton A. Muravev, Tatiana V. Sviridova, Vadim P. Boyarskiy, Andrei L. Kholkin, Ekaterina V. Skorb

**Affiliations:** 1Infochemistry Scientific Center, ITMO University, 191002 St. Petersburg, Russia; 2Chemistry Department, Belarussian State University, 220030 Minsk, Belarus; 3Institute of Chemistry, St. Petersburg State University, 198504 St. Petersburg, Russia; 4Institute of Natural Sciences and Mathematics, Ural Federal University, 260026 Yekaterinburg, Russia

**Keywords:** Knoevenagel condensation, vanillin derivatives, barbituric acid, piezoelectric effect, DFT calculations

## Abstract

A concept of piezo-responsive hydrogen-bonded π-π-stacked organic frameworks made from Knoevenagel-condensed vanillin–barbiturate conjugates was proposed. Replacement of the substituent at the ether oxygen atom of the vanillin moiety from methyl (compound **3a**) to ethyl (compound **3b**) changed the appearance of the products from rigid rods to porous structures according to optical microscopy and scanning electron microscopy (SEM), and led to a decrease in the degree of crystallinity of corresponding powders according to X-ray diffractometry (XRD). Quantum chemical calculations of possible dimer models of vanillin–barbiturate conjugates using density functional theory (DFT) revealed that π-π stacking between aryl rings of the vanillin moiety stabilized the dimer to a greater extent than hydrogen bonding between carbonyl oxygen atoms and amide hydrogen atoms. According to piezoresponse force microscopy (PFM), there was a notable decrease in the vertical piezo-coefficient upon transition from rigid rods of compound **3a** to irregular-shaped aggregates of compound **3b** (average values of *d*_33_ coefficient corresponded to 2.74 ± 0.54 pm/V and 0.57 ± 0.11 pm/V), which is comparable to that of lithium niobate (*d*_33_ coefficient was 7 pm/V).

## 1. Introduction

Piezoelectric response is a useful figure of merit for energy storage, catalysis, and detection of ionic and molecular analytes. Its advantages include quick response time, easy integrity with electrochemical equipment, and low detection limit [1,2,3,4,5,6].

Inorganic salts and oxides (quartz, ZrO_2_, ZnO) are conventional piezomaterials. These are, however, quite dependent on film-fabrication techniques and their morphology is thus rarely reproducible [7]. Regarding biological applications, transition metal oxides are regarded as environmentally hazardous and there are yet few receptor molecules with an effective binding capacity towards biorelevant substrates, which display a significant piezoelectric effect [8]. An interesting approach has been very recently suggested by a group of authors who employed amino acids for the development of crystalline materials with a piezoelectric effect comparable to that of barium titanate [9,10,11]. Amino acids can readily form hydrogen-bonded organic frameworks, but supramolecular organization of piezomaterials based on amino acids on the substrate was not studied except for the crystal state. This points that the noncovalent assembly of organic compounds could lead to the discovery of new materials with record-high piezoresponse and improved sensing features.

Knowing that barbiturate derivatives self-assemble into rosette- and tape-like frameworks [12,13,14] and vanillin is an easily accessible water-soluble molecule, we designed and synthesized vanillin–barbiturate conjugates and investigated their self-assembly pattern and piezoelectric behavior.

## 2. Results and Conclusions

We suggested two derivatives of vanillin moiety with a different steric environment of phenolic OH group for the CH-functionalization of barbituric acid: vanillin **1a** and ethylvanillin **1b**. Knoevenagel reaction between barbituric acid and vanillin derivatives under mild conditions (at room temperature) afforded the condensation products **3a,b** with a newly formed C=C bond at C-5 position of barbiturate motif within minutes, which was easily controlled by the formation of a yellow colloidal precipitate (Figure 1). The structure of the compounds was unambiguously confirmed by ^1^H and ^13^C NMR spectra (Appendix A), which agree well with previously published physical data [15,16].

Visualization of the Knoevenagel condensation between barbituric acid and vanillin derivatives in water by optical microscopy, as outlined in Appendix A, revealed the formation of needle-shaped crystals in compounds **3a** and **3b** (Figure 1a,c). Interestingly, the precipitate in compound **3b** was also represented by bundle aggregates and one can assign these structures to the amorphous phase due to the absence of attenuation of polarized light (Appendix A) by these bundles. Scanning electron microscopy of the precipitate filtered out from the reaction mixture and washed with ethyl acetate and water (to avoid the interference from unreacted vanillin and barbituric acid) at a 4000–40,000× zoom on the surface of gold-coated aluminum-foil-supported optical microscopy observations and revealed rod-shaped aggregates with the cross-section of ca. 1 × 1 μm^2^ in case of compound **3a**, and more curved, irregular-shaped aggregates were detected in ethyl-functionalized derivative **3b** (Figure 1b,d).

The totality of optical and electron microscopy data suggests a different phase behavior of vanillin–barbiturate conjugates **3a** and **3b**. Indeed, there was a remarkable difference in the crystallinity of the powders, as deduced from powder X-ray diffractograms (Figure 1e) (87% of crystallinity for methoxy-functionalized compound **3a** and 57% for ethyl counterpart **3b**). Nevertheless, similar diffraction patterns of vanillin–barbiturate conjugates according to the peak number and their position in the 2θ range of 3–30° indicate their analogous self-assembly in solid state, in particular, the reflexes between 2θ values of 25.5° and 27.9°, which presumably indicate the planar distances between aryl rings (3.2–3.5 Å). Surprisingly, the reflexes in the previously published theoretical powder diffractogram of the single crystals of compound **3a** [15] (see Appendix A therein) did not coincide with those in the experimental spectrum in this work. In particular, no reflexes were previously reported at 2θ = 5.57° corresponding to the spacing of 1.59 nm between the diffracting planes. Such difference may originate from different conditions of synthesis, which could affect the phase state of compounds **3a** and **3b** in the powders (synthesis in aqueous solution used in this work and mechanochemical synthesis in [15]). These data suggest a different self-assembly pattern of compound **3a** in solid state from the one published previously.

Theoretical calculations were further carried out using density functional theory (DFT) to rationalize the assembly mode of compounds **3a** and **3b** and evaluate the thermodynamic favorability of the formation of dimeric supramolecular associates in the initial stages of self-assembly (Figure 2) (computational details section in Supporting Information and references therein [17,18]).

We proposed three hydrogen-bonded dimer models, which resemble the fragment of crinkled tape (**1′** and **1″**), linear tape (**2′** and **2″**), and rosette (**3′** and **3″**) assemblies. The calculations of their Δ*G* values showed that both compounds are mostly stabilized at the H-bonding between carbonyl O atoms of barbituric acid at the *ortho* position to exocyclic C=C double bond and NH hydrogen atoms (by 18–25 kJ/mol in terms of Gibbs free energies, dimers **2′** and **2″**) and this result agrees well with the H-bonding pattern observed in single crystals of compound **3a** given in the literature [15]. Interestingly, verification of the linear tape H-bonded assembly by geometry optimization of the trimer obtained by attaching the monomer molecule to dimer **2″** revealed a partial transformation from linear to stacked configuration, with face-to-face orientation of barbituric acid moieties and the distance of ca. 3.9 Å (Appendix A: transformation from initial geometry to optimized geometry). Therefore, contribution of stacking interactions between barbituric acid moieties was evaluated in dimers **6′** and **6″** and was shown to significantly stabilize the molecule relative to dimers **2′** and **2″**, respectively (by ca. 15–25 kJ/mol in terms of Gibbs free energies). Account for the stacking arrangement analogous to the crystals of compound **3a** [15], with proximally located vanillin and barbituric acid fragments and opposite orientation of aryl ether fragments (dimers **4′** and **4″**), showed nearly the same effect. Surprisingly, initial geometry of the dimers with close arrangement of vanillin moieties collapsed during the geometry optimization procedure into π-π-stacked vanillin–barbiturate dimers **5′** and **5″** with co-directed aryl ether functionalities (Appendix A: transformation from initial geometry to optimized geometry). Moreover, the latter dimers demonstrated the most favorable stacking interactions (stabilization by up to 50 kJ/mol). This fact is due to the shorter distance between the stacked fragments (3.4–3.6 Å).

Low-frequency piezoresponse force microscopy (PFM) study in contact mode (corresponding force–distance curves are given in Appendix A) of the powders of compounds **3a** and **3b** transferred to aluminum foil revealed that the average amplitude of the piezo signal from photodetector was much higher in the case of compound **3a** as compared with compound **3b** (Figure 3) (full experimental details are given in Materials and Methods section and references therein [19,20]). Conversion of the vertical PFM signal values into the *d*_33_ coefficient provided the average values of 2.74 ± 0.54 (compound **3a**) and 0.57 ± 0.11 pm/V (compound **3b**). To account for the interference of scanning artefacts and background signals (influenced by adhesion force of crystals to probe, local humidity near surface, degree of adhesion of specimen to probe, thermal drifts, moisture layer on specimen, and surface charge), a total of three point measurements were carried out from the same area, which was close to those recorded from the area (2.52 ± 0.49 (point 1), 1.93 ± 0.12 (point 2), and 4.64 ± 0.49 pm/V (point 3) for compound **3a** and 1.11 ± 0.29 (point 1), 1.29 ± 0.36 (point 2), and 0.91 ± 0.04 pm/V (point 3) for compound **3b**). A large dispersion in piezoelectric coefficient values for compound **3a** can be rationalized by a different arrangement of rigid rods and, consequently, different polarization vector relative to the AFM probe. The obtained values of piezoelectric coefficient are comparable to those of amino acids [9] and lithium niobate (7 pm/V), which was used as reference sample in this work.

Thus, vanillin–barbiturate conjugates were readily prepared through Knoevenagel condensation under ambient conditions in water. Theoretical calculations revealed a significant gain in the Gibbs energy of dimerization of the conjugates due to their stacking arrangement, whereas hydrogen bonding contributed to a lower extent to the overall stabilization of the dimers. Relatively high piezoelectric coefficients of the barbituric acid–vanillin conjugates offer their potential as piezo-sensors and generators, biocompatible scaffolds for tissue engineering, and energy harvesters.

## 3. Materials and Methods

*Synthesis*. The precursors (4-hydroxy-3-methoxy-benzaldehyde/4-hydroxy-3-ethoxy-benzaldehyde and barbituric acid) were dissolved separately in water at room temperature under ultrasonication. The solutions were combined and the precipitate was filtered after 24 hours, washed with water and ethyl acetate, and dried. ^1^H NMR spectra were recorded at *T* = 298 K on a 400 MHz Bruker Avance III spectrometer in DMSO-d_6_ as an internal reference (2.50 ppm for ^1^H NMR). ^13^C NMR spectra were recorded on a 101 MHz Bruker Avance III spectrometer in DMSO-d_6_ as an internal reference (39.5 ppm for ^13^C NMR).

*Optical microscopy*. Two 5-μL droplets of 20 mM individual solutions of compound **3a** and compound **3b** and barbituric acid were mixed on the glass slide and dried in the air (Appendix A). Optical images were recorded by LeicaDMI8 optical microscope in transmission mode. A 40× objective was used for image acquisition. Following acquisition parameters were used: exposure time is 100.22 ms, light intensity is 35, aperture f/16 for bright field mode, and under polarized light.

*Powder X-ray diffraction*. The microcrystals of compounds **3a** and **3b** were ground using mortar and pestle and transferred to a plastic cuvette to form a layer that is 2 mm in thickness. Powder X-ray diffraction data were collected using a D2 Phaser diffractometer (Bruker, Germany) with Cu *K*_α_ radiation (1.5406 Å).

*Scanning electron microscopy*. The powder of compound **3a** and **3b** was suspended in 95% EtOH that the concentration of the solution was 1 mM. Sample were deposited onto aluminum, with gold coating for better conductivity. An Inspect scanning electron microscope (FEI, USA) was used to study the microstructure of samples. Measurements were performed under the pressure of 10^−3^–10^−4^ Pa and an accelerating voltage of 20 kV in the mode of secondary electron detection.

*Computational details*. Full geometry optimization for all model structures was carried out at the B3LYP-D3/def2-SVP level of theory with the help of the Orca 4.2.1 program package [17]. The RIJCOSX approximation [18] utilizing def2-SVP/C auxiliary basis set and spin restricted approximation were employed. The convergence tolerances for the geometry optimization were energy change = 1.0 × 10^−6^ Eh, maximal gradient = 1.0 × 10^−4^ Eh/Bohr, RMS gradient = 3.0 × 10^−5^ Eh/Bohr, maximal displacement = 1.0 × 10^−3^ Bohr, and RMS displacement = 6.0 × 10^−4^ Bohr. The Hessian matrices were calculated for all optimized model structures in order to prove the location of correct stationary points on the potential energy surfaces (no imaginary frequencies were found in all cases) and to estimate the thermodynamic properties (viz. enthalpy, entropy and Gibbs free energy) for all model systems at 298.15 K and 1 atm. Cartesian coordinates for **3a** was obtained from literature [15]. For Cartesian atomic coordinates of all optimized equilibrium model structures see attached zip-archive.

*Piezoresponse force microscopy*. The samples were deposited onto aluminum foil without gold coating as previously described for the SEM measurements. The morphology and mechanical properties of the surface of powders of compounds **3a** and **3b** were measured in a semi-contact mode on a SmartSPM scanning probe microscope (NT-MDT, Russia) equipped with the probes of HA_NC series (NT-MDT, Russia). Semi-contact mode was chosen as the primary mode to reduce the invasive impact of the probe on the sample surface. Images of the specimens were recorded in air at a temperature of 21–23 °C and relative humidity of 30–40%. Passive and active vibration protection was used to reduce vibration noise. The Optem optical system with an 85×–1050× zoom camera was used for probe alignment and selection of the area for PFM measurements. The amplitude and phase of the vertical piezoelectric oscillations were recorded by a lock-in amplifier in in the contact mode. The amplitude of the oscillations provides information about the magnitude of the piezoelectric response; and the phase, about the direction of polarization. Piezoelectric response measurements were carried out both from the entire image and in local areas in the image. Image processing was carried out with the Gwyddion program [20]. The average value of the piezoelectric response was calculated from a set of individual values of image points (pixels) as the arithmetic average from all image points of the locally selected area. To measure the local piezoelectric response, the average value of the force constant of the probe was selected in the range of 10–12 N/m to reduce the influence of electrostatic contribution on measurements. The tip radius of the probe according to the certified values was about 35 nm. The piezoelectric coefficient was calculated as follows:$$d_{33}=\frac{D\left(a.u.\mathrm{ADC}\right)Slope\left(\mathrm{pm}/a.u.\mathrm{ADC}\right)}{Q\left(a.u.\right)U\left(V\right)}\;P,$$
where *D* is the value of the output signal from the photodiode (a.u. ADC (analog-to-digital converter)), Slope is the value of descending branch of the force-distance curve conversion of a.u. to picometers through the corresponding curves (Appendix A), *Q* is the measured signal (a.u.), *U* is the applied bias voltage between the probe and the sample (V), and *P* is the adjustable coefficient, which represents the ratio of the measured piezoelectric response of the lithium niobate single crystal (LiNbO_3_) to its reference value (7 pm/V). To reduce the contribution of the electrostatic response on piezoelectric measurements, its operating frequency was taken as 20 kHz (much less than the contact resonance frequency).

## Data Availability

The data presented in this study are available in article and Appendix A.

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
