# Peer review of "Piezo-Responsive Hydrogen-Bonded Frameworks Based on Vanillin-Barbiturate Conjugates"

_molecules, 2022, doi:10.3390/molecules27175659_

Round 1

Reviewer 1 Report

This article proposes a new concept of piezo-responsive hydrogen-bonded π-π stacked organic frameworks. The researchers designed and synthesized vanillin–barbiturate conjugates and investigated their self-assembly pattern and piezoelectric behavior. Some comments and suggestions are given as below.

1. Some main experimental results are suggested to be presented in Abstract.

2. The author indicated that “the reflexes in the previously published theoretical powder diffractogram of the single crystals of compound 3a did not coincide with those in the experimental spectrum in present work.” Please give more discussion on this phenomenon (e.g. specific conditions of synthesis and experimental characterization).

3. The author indicated that “To account for the interference of scanning artefacts and background signals, a total of three point measurements were carried out from the same area, which was close to those recorded from the area (2.52±0.49 (point 1), 1.93±0.12 (point 2), and 4.64±0.49 pm/V (point 3) for compound 3a and 1.11±0.29 (point 1), 1.29±0.36 (point 2), and 0.91±0.04 pm/V (point 3) for compound 3b).” It appears that for compound 3a, the three point measurement results are somewhat discrete, the result for point 3 (4.64±0.49 pm/V) is about 2 times larger than point 2 (1.93±0.12 (point 2)), which needs a reasonable explanation.

Author Response

The authors thank the Reviewer for careful reading of the manuscript text and comments.

In accordance with the Reviewer's suggestions, main experimental results were added to the Abstract (lines 20-29).

Regarding the comment on different powder diffractogram data in this work and those published previously for compound 3a ("give more discussion on this phenomenon (e.g. specific conditions of synthesis and experimental characterization)"), origin of their difference was discussed in the text in context of different synthesis conditions (lines 91-93) and more discussion was added to lines 85-91. This explanation is reasonable, because the phase state of powders could be sensitive to the presence of solvent molecules (in this work, water as reaction medium and ethylacetate that was used to wash the precipitate) and mechanical grinding (ref. 15); more detail was added to the text (line 91). Lines 118-126 also refer to the previously published crystal structure of compound 3a in context of difference of supramolecular association within dimer models (proximal location of vanillin and barbituric acid fragments in dimer 4' (the same as in the crystal of compound 3a [15]) and close arrangement of aryl ether functionalities in dimer 5', which is energetically the most favorable orientation in dimer models). 

In response to the third comment, a large dispersion in piezoelectric coefficient values for compound can be rationalized by a different arrangement of rigid rods and, consequently, different polarization vector relative to the AFM probe (this was added to the text, lines 140-143). In this case, the position of probe and applied external electric field E during application of alternating voltage on the probe-specimen interface are always unambiguous. Interaction of the polarization vector of a particular crystal with external electric field E ultimately determines the value of inverse piezoelectric response represented by the amplitude along Z axis. In addition, the results of measurements could also be affected by adhesion force of crystals to probe, local humidity near surface, degree of adhesion of specimen to probe, thermal drifts, moisture layer on specimen, surface charge (this was added to the text, lines 135-137), and other factors, because the procedure considers vibrational and contact interactions between probe and specimen. Therefore, the piezoelectric coefficient averaged by several measurement points (given in the manuscript) needs to be considered for a valid comparison between different specimens.

Reviewer 2 Report

Report on the paper by Nebalueva et al.

It is an interesting paper on piezo-response of vanillin-barbiturate conjugates. However, there are some unclear points.

1. In line 118-119, it seems that the magnitude of the force is larger for 3b than that for 3a according to Figure S7 in contrary to the description in the manuscript. Then, it is required to explain in more details in the manuscript why d33 of 3a is much larger than that of 3b in spite of the fact that the magnitude of the force for 3b was larger than that for 3a in Fig. 7S.

2. In line 57, does Figure S5 mean Figure S6?

3. In line 58, (Figure 1 a , b) should be corrected as (Figure 1 a, c).

4. In line 61, the experimental data of the attenuation of polarized light is absent in the supplementary material. Figure S6 is the optical microscope images.

5. In line 67, (Figure 1c, d) should be corrected as (Figure 1 b, d).

6. In line 69 (the caption of Figure 1), (a, b) should be corrected as (a,c). (c, d) should be corrected as (b, d). (a, c) should be corrected as (a, b). (b, d) should be corrected as (c, d).

7. Figures S1-S4 are not referred in the manuscript.   

Author Response

The authors thank the Reviewer for careful reading of the manuscript text and comments. A point-by-point response is given below.

  1. Figure S7 shows the force curves of cantilever deflection during contact-mode interaction of probe with specimen ( and 3b). These curves are recorded before measurement of piezoelectric coefficient and they are used to re-calculate indications of instrument from [nA], which is the signal of cantilever deflection relative to the center of photodiode, into [nm], which is the real cantilever deflection value. The difference in signals considers the degree of stiffness of probe-specimen contact for specimens and 3b. The d33 coefficient was larger in the case of compound 3a than that of compound 3b, because the vertical amplitude during measurement of inverse piezoelectric effect was larger in that case.
  2. No, there is a reference to Figure S5 in line 57 rather than Figure S6.
  3. Corrected.
  4. Figure S6 shows polarized optical microscopy images of compounds 3a and 3b. A brighter image for compound 3b indicates lower degree of attenuation of polarized light as compared to a darker image for compound 3a.
  5. Corrected.
  6. This was corrected.
  7. Reference to Figures S1-S4 is given in line 61.